**Titolo**

**ProgrEFR: diffondere e promuovere i programmi di ricerca dell'Ecole française de Rome con Wikidata /** Elena Avellino, Elisa Saltetto

**ProgrEFR: Disseminating and promoting the research programmes of the Ecole française de Rome with Wikidata**

**Autori**

Elena Avellino

Elisa Saltetto

**Affiliazione**

Bibliothèque de l'Ecole française de Rome

**Parole chiave:**

Ecole française de Rome – programmi di ricerca – Linked Open Data – open science – open data – IdRef –  WikiData – HAL – rete universitaria –

**Biografia Autori**

**Elena Avellino**, dopo aver svolto attività di ricerca e di insegnamento in archeologia preistorica all'Università di Paris X, si è dedicata alla gestione documentale e bibliografica per diverse istituzioni scientifiche e culturali in Francia e in Italia (Maison d'Archéologie et Ethnologie René-Ginouvès, Sous-direction de l'archéologie, Musée des Antiquités Nationales, Biblioteca Baldini). Attualmente è bibliotecaria all'Ecole française de Rome dove si occupa in particolar modo di gestione delle autorithy e di progetti legati ai linked data e all'open science.

**Elena Avellino**, after carrying out research and teaching in prehistoric archaeology at the University of Paris X, she has been involved in documentary and bibliographical management for several scientific and cultural institutions in France and Italy (Maison d'Archéologie et Ethnologie René-Ginouvès, Sous-direction de l'archéologie, Musée des Antiquités Nationales, Biblioteca Baldini). She is currently a librarian at the Ecole française de Rome. She is particularly involved in authority management and projects related to linked data and open science.

**Elena Avellino**, après avoir mené des activités de recherche et d'enseignement en archéologie préhistorique à l'Université de Paris X, elle a travaillé dans la gestion documentaire et bibliographique pour diverses institutions scientifiques et culturelles en France et en Italie (Maison d'Archéologie et Ethnologie René-Ginouvès, Sous-direction de l'archéologie, Musée des Antiquités Nationales, Biblioteca Baldini). Elle est actuellement bibliothécaire à l'Ecole française de Rome. Elle est chargée en particulier de la gestion des autorités et des référentiels et de projets liés à l'interopérabilité entre référentiels et à la science ouverte.

*

**Elisa Saltetto**, filologa di formazione, durante gli studi ha maturato interesse per la professione di bibliotecaria, che ha esercitato principalmente presso sedi italiane di istituzioni estere (Reale Istituto Neerlandese a Roma e Istituto Storico Austriaco a Roma). Attualmente fa parte dell'équipe della biblioteca dell'Ecole française de Rome, dove si occupa in particolare di servizio all'utenza e di gestione delle collezioni, affiancando attivamente i progetti legati allo sviluppo dei *linked open data* e alla scienza aperta.

**Elisa Saltetto**, philologist by training, developed an interest in the profession of librarian during her studies. She worked mainly at Italian branches of foreign institutions (Royal Netherlands Institute in Rome and the Austrian Historical Institute in Rome). She is currently part of the library staff at the Ecole française de Rome, where she is particularly involved in user service and collection management, actively supporting projects related to the development of linked open data and open science.

**Elisa Saltetto**, philologue de formation, s'est intéressée à la profession de bibliothécaire au cours de ses études. Elle a travaillé principalement pour les sièges italiennes de différentes institutions étrangères (Royal Institut néerlandais à Rome et Institut historique autrichien à Rome). Elle fait actuellement partie de l'équipe de la bibliothèque de l'Ecole française de Rome, où elle est particulièrement impliquée dans le service aux usagers et dans la gestion des collections, tout en soutenant des projets liés au développement des *linked open data* et de la science ouverte.

## Abstract

Il nostro progetto mira a

- promuovere e dare visibilità alle attività scientifiche e ai programmi di ricerca della nostra istituzione
- creare dei link tra contenitori di dati di natura complementare (LOD)
- rendere accessibili le risorse prodotte dai ricercatori dell'EFR in un'ottica di open data

Abbiamo scelto di concentrarci su venti programmi di ricerca, detti strutturanti, che costituiscono una parte trainante dell'attuale attività scientifica dell'EFR. Sono articolati in diverse tematiche di ricerca, ricoprono un ampio arco cronologico (dall'antichità alla storia contemporanea) e, attingendo a diverse metodologie e fonti di ricerca, adottano un approccio multidisciplinare. I programmi strutturanti hanno una durata di quattro o cinque anni. Dato il carattere internazionale dell'istituzione, essi sono realizzati in partenariato con una o più istituzioni straniere o italiane.

Per raggiungere questo obiettivo il nostro progetto prevede

- la descrizione dei programmi nel database della rete bibliografica delle università francesi (SUDOC/IdRef)
- la descrizione dei programmi in HAL (data base in cui i ricercatori affiliati al Ministero della Ricerca e dell'Insegnamento superiore francese versano la loro produzione scientifica con libertà di renderne accessibile o meno i contenuti)
- l'inserimento e la descrizione dei programmi in Wikidata

Concretamente, per ogni programma sono stati creati gli identificativi IdRef, Hal e gli item Wikidata con i rispettivi rinvii e descrittori adeguati alle caratteristiche proprie ad ogni data base. Gli identificativi propri ad ogni contenitore di dati permettono il dialogo tra i vari sistemi nell'ottica dell'interoperabilità, ricavando dati e informazioni diverse a seconda delle proprietà/ambiti dei vari contenitori. Così IdRef consente essenzialmente una descrizione del programma come un ente autore, Hal raccoglie i relativi studi depositati dai singoli ricercatori e Wikidata diventa soprattutto l'aggregatore che collega le diverse interfacce.

**ProgrEFR: Disseminating and promoting the research programmes of the Ecole française de Rome with Wikidata**

Our project aims to

- promote and give visibility to the scientific activities and research programmes of our institution

- create links between data containers of a complementary nature (LOD)

- make the resources produced by EFR researchers accessible in an open data perspective

We have chosen to focus on twenty research programmes, known as structuring programmes, which form a driving force in EFR's current scientific activity. They are divided into different research themes, cover a broad chronological span (from antiquity to contemporary history) and, drawing on different research methodologies and sources, adopt a multidisciplinary approach. The structuring programmes have a duration of four or five years. Given the international character of the institution, they are realised in partnership with one or more foreign or Italian institutions.

To achieve this goal, our project involves

- the description of programmes in the database of the bibliographical network of French universities (SUDOC/IdRef)

- the description of the programmes in HAL (a database in which researchers affiliated to the French Ministry of Research and Higher Education deposit their scientific production with freedom to make its contents accessible or not)

- the entry and description of programmes in Wikidata

Specifically, IdRef, Hal and Wikidata item identifiers were created for each programme, with the respective references and descriptors adapted to the characteristics specific to each data base. The identifiers proper to each data container allow dialogue between the various systems with a view to interoperability, obtaining different data and information depending on the properties/environments of the various containers. Thus IdRef essentially allows a description of the programme as an corporate name heading, Hal collects the relevant studies deposited by individual researchers, and Wikidata becomes above all the aggregator linking the different interfaces.

**ProgrEFR : Diffuser et promouvoir les programmes de recherche de l'Ecole française de Rome avec Wikidata**

Notre projet vise à

- promouvoir et donner de la visibilité aux activités scientifiques et aux programmes de recherche de notre institution

- créer des liens entre des réservoirs de données de nature complémentaire (LOD)

- rendre accessibles les ressources produites par les chercheurs de l'EFR dans une perspective d'open data.

Nous avons choisi de nous concentrer sur une vingtaine de programmes de recherche, dits structurants, qui constituent un des moteurs de l'actuelle activité scientifique de l'EFR. Ils sont divisés en différents thèmes de recherche, ils couvrent un cadre chronologique très étendu (de l'Antiquité à l'histoire contemporaine) et adoptent une approche pluridisciplinaire qui fait appel à des méthodologies et des sources de recherche différentes. Les programmes structurants ont une durée de quatre ou cinq ans. Compte tenu du caractère international de l'institution, ils sont réalisés en partenariat avec une ou plusieurs institutions étrangères ou italiennes.

Pour chaque programme nous avons créé les identifiants IdRef, Hal et Wikidata, avec les respectifs références et descripteurs adaptés aux caractéristiques propres à chaque base de données. Ces identifiants permettent le dialogue entre les différents systèmes pour favoriser l'interopérabilité et présenter des données et des informations différentes en fonction des propriétés et de l'environnement de chaque réservoir. Ainsi, IdRef permet essentiellement de décrire le programme en tant que collectivité/auteur, Hal rassemble les études issues des programmes et déposées par les chercheurs associés et Wikidata devient surtout l'agrégateur qui relie les différentes interfaces.

**Tema**

Valorizzazione dei progetti di ricerca dell'Ecole française de Rome e diffusione dei risultati della ricerca scientifica attraverso Wikidata

**Theme**

Enhancement of Ecole française de Rome research projects and dissemination of scientific research results through Wikidata

**Format of the submission**:

Presentazione 20 min

**Lingua**

Italiano e inglese

**Presenza a Firenze**

Elena Avellino

Elisa Saltetto

# License CC BY 4.0, CC BY-SA 4.0

Authorization to use personal data for the purpose of the conference organization by Wikimedia Italia according to its privacy policy.

We accept the Wikimedia Universal code of conduct and Wikimedia friendly space policy

