# OpenReview forum: "ProgrEFR: diffondere e promuovere i programmi di ricerca dell’Ecole française de Rome con Wikidata"
_wikimedia.it/Wikidata_and_Research/2025/Conference — WD&R LT_

### Official Review · ~Lucia_Sardo1 · 2025-01-07
**revisione**

**Originality:** 5
**Impact:** 4
**Confidence:** 4

**Review:**

La proposta risulta ben strutturata, con obiettivi di progetto e metodologia impiegata chiari e ben presentati. La realizzazione di questo progetto potrebbe avere senza dubbio un impatto positivo per la conoscenza delle attività dell'EFR e per la promozione di progetti simili presso altre istituzioni simili. Dall'abstract però non è possibile ricavare la struttura dei dati e  la modalità operativa che sarà seguita per la realizzazione del progetto stesso.

**Compliance:**

4

**Scientific Quality:**

3

---

### Official Review · ~Rossana_Morriello1 · 2025-01-07
**Programmi e temi di ricerca dell'EFR in Wikidata**

**Originality:** 4
**Impact:** 3
**Confidence:** 3

**Review:**

Il progetto è chiaro e abbastanza interessante, nonostante la perplessità sulla breve durata dei programmi (4-5 anni dopo i quali vengono aggiornati regolarmente? in che modo?) e l'impatto sulla comunità Wikidata, al di là delle istituzioni coinvolte nei programmi. Due aspetti sui quali suggerirei di soffermarsi.

**Compliance:**

4

**Scientific Quality:**

3

---

### Decision · Program_Chairs · 2025-02-05

**Decision:**

Accept (LT)

**Comment:**

== Scelto il LT ==

Dear Authors,
thank you very much for your proposal. We regret to inform you that your proposal was not selected among the papers.

Even if not selected as paper, we consider your proposal relevant and interesting and we would like to propose you to prepare instead a lightening talk (if you - or another member of your team - can participate in presence at the conference) or a poster (which can be exhibited even if you will not attend the conference).

It would be a pleasure to learn more about your work through a lightening talk or a poster.
Thank you for submitting a proposal and please let us know if you like the idea of converting it into a lightening talk or a poster and which format you prefer.

Regards,
The scientific committee of the conference Wikidata and Research